# Educational Videos to Reduce Parental Rejection of Pediatric Cardiac Catheterization during the COVID-19 Pandemic

**DOI:** 10.3390/healthcare11101395

**Published:** 2023-05-11

**Authors:** Shu-Juan Liu, Yan-Zin Chang, Lien-Jen Hwu, Min-Sho Ku

**Affiliations:** 1Institute of Medicine, Chung Shan Medical University, Taichung 40201, Taiwan; 2Department of Nursing, Chung Shan Medical University, Taichung 40201, Taiwan; 3Department of Nursing, Chung Shan Medical University Hospital, Taichung 402367, Taiwan; 4Drug Testing Center, Department of Clinical Laboratory, Chung Shan Medical University Hospital, Taichung 402367, Taiwan; 5School of Medicine, Chung Shan Medical University, Taichung 40201, Taiwan; 6Division of Allergy, Asthma and Rheumatology, Department of Pediatrics, Chung Shan Medical University Hospital, Taichung 402367, Taiwan

**Keywords:** COVID-19, parental uncertainty, anxiety, educational video, cardiac catheterization

## Abstract

During the COVID-19 pandemic, people voluntarily reduced their necessary healthcare. We examined whether supplying educational digital versatile discs (DVDs) before admission can reduce parental rejection of pediatric cardiac catheterization for congenital heart disease (CHD). Parents of 70 children with CHD selected for cardiac catheterization were randomly allocated to the DVD (received pre-admission DVDs in the outpatient department; 70 parents of 35 children) or non-DVD groups (did not receive the DVDs; 70 parents of 35 children). The parents could reject the admission of their children within 7 days. Cardiac catheterization was rejected by 14 (20.0%) and 26 (37.1%) parents in the DVD and non-DVD groups, respectively (*p* = 0.025). Parent Perceptions of Uncertainty Scale scores were lower in the DVD (128.3 ± 8.9 points) than in the non-DVD group (134.1 ± 7.3 points; *p* < 0.001). Decreased uncertainty due to pre-admission DVD watching could have contributed to the increased parental willingness for cardiac catheterization. The effects of pre-admission educational DVDs were more significant among parents with a lower education, rural residence, with only one child, female child, or younger child. Offering educational DVDs to parents of children selected for cardiac catheterization for CHD may decrease the parental rejection rate of the treatment.

## 1. Introduction

The COVID-19 pandemic has considerably altered people’s lives. It prevented people from keeping regular medical appointments even in an area with a low infection risk [1] and resulted in reduced admissions to hospital and emergency department visits [2]. The number of outpatient visits also largely decreased [3]. The condition was observed in Taiwan and other countries [1,2,3,4]. The decline in health utilization might impede personal disease control and increase the mortality rate. For example, a German study reported a significant decrease in the number of cardiac catheterization procedures performed during the pandemic and a concomitant increase in the cardiovascular-related mortality rate [4]. They reported that the cardiovascular and cardiac mortality rates increased significantly by 7.6% and 11.8%, respectively. This decrease in seeking medical care was attributed to increased anxiety, intolerance of uncertainty, and fear of death [5,6].

Studies have indicated that parents who had children with chronic illnesses or children who underwent surgical procedures presented increased uncertainty and anxiety. Increased parental uncertainty and anxiety can have several adverse effects. For example, they would decrease the quality of care for their sick child [7]. They can also impact psychological and physical symptoms in mothers [8], parents’ distress, and children’s depressive symptoms [9], and interfere with recovery from surgical procedures [10]. Parents of children with congenital heart disease (CHD) selected to undergo cardiac treatment (surgery or catheterization) also experience increased uncertainty and anxiety due to the fear of morbidity or mortality caused by the treatment. Given the aggravation of parental uncertainty and anxiety during the COVID-19 pandemic, the aforementioned adverse effects are expected to have also increased.

Therefore, there is a need to establish interventions for reducing parental uncertainty and anxiety to mitigate the related adverse effects. Providing adequate and efficient information to parents has been reported to decrease their uncertainty and anxiety [11]. Such benefit was also noted when their children underwent surgery for congenital heart disease [12]. An evidence-based project demonstrates that for patients whose children received surgical treatment, adding digital versatile disc (DVD) education to routine education is helpful for distributing important information [13]. The use of DVDs to prepare patients for the cardiac catheterization is effective in reducing the level of anxiety and the uncertainty, as the patients experience higher satisfaction and knowledge level after the educational intervention [14]. Pre-anesthesia educational DVDs can also reduce parental anxiety and desire for information before pediatric day-case surgery [15]. The study results indicated that, in cardiac catheterization or surgery, educational DVDs effectively informed parents and patients, which helped decrease their uncertainty and anxiety.

Since March 2022, the Taiwanese government has decided to seek ways to cope with, rather than eradicate, COVID-19; moreover, the number of COVID-19 cases has dramatically increased [16]. Accordingly, there has been a decrease in individuals seeking medical care for diseases other than COVID-19 due to fear [1,3]. We arranged a study during the period of the COVID-19 pandemic. For parents whose children were informed about cardiac catheterization treatment for CHD in the outpatient department, a pre-admission educational DVD was given to parents, which they watched at an outpatient department and at home. We aimed to know whether or not the pre-admission DVD offered in the outpatient department can decrease parental rejection for admission for cardiac catheterization during the COVID-19 panic. In parents whose children were admitted for cardiac catheterization, the same DVD was also added to the routine education. We aimed to examine whether or not the addition of DVD education can decrease parental uncertainty and anxiety during admission. 

## 2. Materials and Methods 

### 2.1. Design

Figure 1 shows the study flow chart. A single blind, randomized case–control study was conducted.

### 2.2. Participants

Between 27 July 2022 and 26 January 2023, 70 children aged 3–12 years underwent elective and necessary cardiac catheterization treatment for patent ductus arteriosus (PDA), atrial septal defect (ASD), or ventricular septal defect (VSD). The treatment included trans-catheter closure of PDA, trans-catheter closure of ASD, and trans-catheter closure of VSD. We did not consider children selected for emergent or non-elective catheterization. Additionally, we excluded children with unstable vital signs or those requiring a high drug dose to preserve heart function. Furthermore, we excluded children with genetic disorders, non-cardiac congenital anomalies, chronic diseases (e.g., cerebral palsy, epilepsy, psychiatric diseases, and chronic lung diseases), and a history of surgical treatment. We included parents who are the primary caregivers and could effectively communicate either verbally or in writing. Additionally, we excluded parents with emotional or mental disorders (e.g., depression and anxiety) or severe diseases (e.g., cancer). Finally, we excluded unmarried fathers, mothers, and single parents (either divorced or widowed) given the potential influence of their emotional conditions. 

### 2.3. Group Allocation and Education in the Outpatient Department

Parents were randomly divided into the DVD and non-DVD groups using computer-generated random numbers. At 7 days before admission, in the outpatient department of the attending physician, parents in the DVD group received oral information regarding cardiac catheterization and brief routine information on the treatment, procedure, and precautions taken for COVID-19. Moreover, they received a pre-admission educational DVD, which could be watched by parents, children, or other family members at home. Parents in the non-DVD group only received the aforementioned oral information and brief routine education. Viewing of the DVD and the brief routine information were carried out in the outpatient department, with only the parents, child, physicians, and nurses who arranged the study. During the subsequent 7 days, the parents decided whether to accept or reject admission for pediatric cardiac catheterization treatment. 

### 2.4. Educational DVD

A 10-min educational DVD was created by the authors and hospital staff. It included information regarding the cardiac catheterization procedures, the intensive care unit (ICU) environment and equipment (including when the equipment was used and removed), post-catheterization care (including drug usage and feeding), and catheterization or post-catheterization complications. Moreover, it included a simulation of a real case from admission to discharge. Additionally, it included information regarding peri-admission prevention strategies for COVID-19 infection, treatment interventions for COVID-19 infection, strategies for preventing related complications or death, and how patient visits could be made through video calls. The DVD was reviewed by two pediatric cardiologists and three clinical nurses (with PhDs) as well as three fathers and mothers without a medical background. The content validity index was 0.92. Their suggestions were used to modify the DVD content accordingly to allow easy comprehension. As no major deficiencies were observed in the pilot test, no changes were made in the subsequent studies. 

### 2.5. Education after Admission

On the day of admission, parents in the DVD group received routine education and re-watched the educational DVD. Contrastingly, parents in the non-DVD group only received routine education, which comprised similar content as the educational DVD but was verbally delivered by doctors and nurses with the aid of educational leaflets. All parents were allowed to ask questions during the education. Experienced and trained physicians and nurses of the same team, who arranged the study, delivered the information verbally, obtained parents’ consent, and recorded the data. They also delivered the routine education and offered the DVD to the parents. The nurses and physicians who actually cared for the patients during admission were not involved in administering the educational protocol and were completely blinded to the study.

### 2.6. Instruments 

The Chinese versions of the Beck Anxiety Inventory (BAI) [17] and Mishel’s Parent Perceptions of Uncertainty Scale (PPUS) [18] were used to evaluate parental anxiety and uncertainty levels, respectively. The BAI comprises 21 questions rated on a 4-point scale, while the PPUS comprises 31 items rated on a 5-point scale. High total scores indicated high levels of anxiety or uncertainty. Both scales have been shown to have good reliability and validity. Cronbach’s α values for the PPUS and BAI were 0.91 [18] and 0.95 [19], respectively. After admission, viewing of the DVD, routine education, and BAI and PPUS tests were carried out in the education room of the pediatric ICU of the hospital. Only the parents, child, physicians, and nurses who arranged the study participated in the course.

### 2.7. Data Collection and Between-Group Comparison

After admission, routine education, viewing of the DVD, and the PPUS and BAI tests were performed in the education room of the pediatric ICU. The fathers and mothers separately completed both tests on the admission day before and at 4 h after receiving the education as well as on the discharge day, with guidance from the same nurse for each pair of parents. 

### 2.8. Confounding Factors 

We adjusted for confounding factors that may influence the level of parental uncertainty or anxiety [20,21]. These confounding factors included parental age, sex, and education (high: equal or higher than university level; low: equal or lower than junior college level); family income (high: family income more than USD 30,000 per year; low: family income equal or less than USD 30,000 per year); place of residence (urban or rural); presence of siblings; and children’s age, sex, and heart disease type (PDA, ASD, or VSD).

### 2.9. Ethical Considerations

The institutional review board of Chung Shan Medical University Hospital, Taichung, Taiwan, approved this epidemiological study (ethics approval number: CS2-22078). All parents provided written informed consent 7 days before admission. Demographic information was collected from the parents on the same day. They were informed that they could withdraw from the study at any point (before or during admission) and that the quality of care provided would not be influenced by their refusal to participate or withdraw from the study.

### 2.10. Statistical Analyses

All statistical analyses were performed using Predictive Analytics Software Statistics 18 (IBM Corp., Armonk, NY, USA). The chi-square test was used for between-group comparisons of categorical variables (demographic data and rate of rejection for cardiac catheterization). An independent *t*-test was conducted for between-group comparisons of categorical variables (demographic data, PPUS scores, and BAI scores). Confounding factors were adjusted using multivariate regression analysis. Two-sided *p*-values < 0.05 were defined as statistically significant.

## 3. Results

### 3.1. Demographic Data 

Table 1 presents the demographic data. Each group comprised 70 parents (35 fathers and 35 mothers) of 35 children. There were no significant between-group differences in the demographic characteristics.

### 3.2. The Rejection Rate for Cardiac Catheterization for CHD

In the DVD and non-DVD groups, 14 (20.0%) and 26 (37.1%) parents rejected the cardiac catheterization treatment, respectively (Table 2). This indicated a higher rejection rate in the non-DVD group than in the DVD group (odds ratio [OR], 0.42; 95% confidence interval [CI], 0.20–0.91; *p* = 0.025). 

### 3.3. Factors That Influence the Rejection Rate for Cardiac Catheterization for CHD

Table 2 presents the factors that influenced the rejection rates for cardiac catheterization for CHD. Among parents with lower educational levels, the rejection rate was lower in the DVD group than in the non-DVD group (24.4% vs. 46.2%; OR, 0.38; 95% CI, 0.15–0.97; *p* = 0.041); however, this between-group difference was not observed among parents with a higher educational level. Moreover, compared with the non-DVD group, the DVD group showed a specifically lower rejection rate among parents with rural residence, only one child, a female child, or a younger child. 

### 3.4. Uncertainty Scores at Different Time Points

None of the admitted pediatric patients showed major complications, including tracheal intubation, prolonged admission, severe infection, and severe bleeding. Furthermore, none of the admitted children was withdrawn from the study. Table 3 shows the mean PPUS scores at the different time points. Before education on the admission day, the mean PPUS scores were lower in the DVD (128.3 ± 8.9) than in the non-DVD group (134.1 ± 7.3; *p* < 0.001). After education on the admission day and on the discharge day, the mean PPUS scores were also significantly lower in the DVD group. 

### 3.5. Anxiety Scores at Different Time Points

Table 4 shows the mean BAI scores at the different time points. Before education on the day of admission, there was no significant between-group difference in the mean BAI score. After education on the admission and discharge days, the mean BAI scores were significantly lower in the DVD group than in the non-DVD group.

## 4. Discussion

Our findings indicated that supplying an educational DVD in the outpatient department 7 days before admission could significantly decrease parental rejection of pediatric cardiac catheterization treatment for CHD during the COVID-19 pandemic. The effects of the pre-admission educational DVD were more significant among parents with a lower education level, who lived in the country, had only one child, had a child of female sex, or had a child of a young age. Decreased parental uncertainty due to DVD education might be the reason for the decreased rejection rate. Furthermore, re-watching the DVD during the routine education on the admission day could decrease parental uncertainty and anxiety compared with only administering routine education. The decreased parental uncertainty and anxiety could be observed after education (before catheterization) and could last until the day of discharge.

Except for human mobility restrictions or the constraint of hospital services, people voluntarily reduced their demand for healthcare due to the fear of COVID-19 [3]. Delay or interruption of conventional treatment to pandemic-related fears may increase the risk of deterioration or reduce the survival rates in patients with cardiovascular disease [4]. Delaying breast, lung, and colon cancer surgeries during the COVID-19 pandemic may decrease their survival rate [22]. Parents of children selected to undergo cardiac procedures for CHD experience high uncertainty and anxiety levels [23]. Worry related to the risk of exposure to COVID-19 during healthcare visits contributes to an increase in parental anxiety [24]. Other factors that may contribute to increased parental uncertainty during the COVID-19 pandemic include uncertainty regarding the follow-up schedule as well as factors related to the complexity and stability of their children’s condition [25]. In our study, we found that supplying a pre-admission educational DVD in the outpatient department could decrease parental uncertainty on the admission day. Therefore, we speculated that, through the reduction in parental uncertainty, pre-admission educational DVD watching could decrease the rejection rates for pediatric cardiac catheterization treatment during the COVID-19 pandemic and panic. 

Uncertainty and anxiety have several adverse effects. Parental uncertainty might adversely affect physiological health in parents [8], increase the depressive symptoms of their children [9], and interfere with their ability to care for their sick children [7]. Parental anxiety can worsen children’s anxiety [26], increase children’s postoperative pain [10], and lead to overprotective behavioral problems [27]. In our study, adding a DVD to the routine education on the day of admission was found to decrease parental uncertainty and anxiety during the COVID-19 pandemic and panic, which might have contributed to preventing the aforementioned adverse effects. 

According to the theory of Mishel [11], uncertainty among parents of hospitalized children may result from (1) an inability to understand the disease progression or treatment plan; (2) inadequate explanations regarding their children’s treatment procedures or care system; and (3) inability to predict the near future of the child’s illness, treatment outcomes, and survival chances. Moreover, inadequate knowledge may also increase the anxiety levels [28]. During the COVID-19 pandemic and panic, parental uncertainty and anxiety must be added. Therefore, it is important to provide adequate information and education to parents in order to decrease parental uncertainty and anxiety. 

Routine education involving verbal instructions and supplemental written materials is the most common format for outpatient and pre-catheterization teaching during admission. The resulting knowledge transfer is influenced by factors such as intellectual level, language or cultural barriers, learning disabilities, and attention span [29]. Educational DVDs have been shown to allow better knowledge transfer as well as decrease the uncertainty and anxiety levels among pediatric patients or their parents [14,30]. A pre-admission supply of educational DVDs can allow improved transfer of information and knowledge. Compared with routine education in the outpatient department, which only occurs once, the DVD can be watched at home, which allows more frequent transmission of information. This could have contributed to the decreased rejection rate in the DVD group. In addition, re-watching the DVD after admission further facilitated the transmission of information. This could explain the decreased parental uncertainty and anxiety levels during admission in the DVD group. 

During the outbreak of the COVID-19 pandemic, a tsunami of disease-related information, which is known as the infodemic, appeared. The infodemic contained accurate and inaccurate information and was transferred from commercial media. Parents without a medical background did not have adequate knowledge to find trustworthy information from the overload of information. This made parents more anxious and uncertain. One study found that the infodemic factors had an adverse impact on public anxiety, and the influence was beyond that of the pandemic factors [31]. Therefore, finding a strategy to transfer accurate information more effectively is crucial during the COVID-19 pandemic. In the study, we examined the efficiency of the following strategy: DVD plus routine education, arranged in the outpatient department and during admission. This educational format can overcome the aforementioned problems. The educational format might also be effective in cases of other epidemic or pandemic diseases in the future.

### 4.1. Limitations

Our study has several limitations. First, it was conducted in a medical center. Selection bias may have occurred because patients who were treated in hospitals of other degrees were not included. Future studies recruiting participants from different hospitals of different degrees are necessary. Second, before admission, parents may have obtained information from commercial media except for the DVD and routine education. Thus, contamination bias may have occurred. Third, parents with an intense fear of COVID-19 may not have brought their children to our cardiac outpatient department and were not gathered in our study. Thus, volunteer bias may have occurred. Fourth, we excluded unmarried, single parents, and parents with psychological or mood disorders. Fifth, we did not include children who underwent catheterization treatment for complicated CHD, including balloon dilation for Tetralogy of Fallot or aorto-pulmonary collateral vessel embolization. These parents and children should be considered in future related studies. Sixth, we did not analyze the effects separately for mothers and fathers owing to the small sample size. Future studies are necessary to examine the influence of our educational format on mother’s or father’s rejection rates for cardiac catheterization treatment and their psychological influence.

### 4.2. Implications for Practice

During the COVID-19 pandemic, people voluntarily reduced their necessary healthcare. This is harmful to their health and disease control and might increase the mortality rates. Information concerning the medical procedures, such as catheterization treatment, is often conducted in the outpatient department. There is no sufficient time to offer adequate education. Inadequate information might increase parental uncertainty and result in their rejection for necessary medical care. The results of our study indicated that in the pediatric cardiac outpatient department, supplying educational DVDs plus routine education to parents at 7 days before admission could decrease parental rejection of pediatric cardiac catheterization treatment for CHD. Such an educational format could decrease parental uncertainty on the day of admission. Moreover, during admission, adding educational DVDs to the routine education before cardiac catheterization could also decrease parental uncertainty and anxiety levels during admission. The benefits could last until the day of discharge.

## 5. Conclusions

During the COVID-19 pandemic, supplying pre-admission educational DVDs in the outpatient department 7 days before admission could decrease parental uncertainty, and consequently decrease rejection of pediatric cardiac catheterization treatment for CHD. Furthermore, adding educational DVDs to the routine education could decrease parental uncertainty and anxiety levels during admission. Except for cardiac catheterization, the educational format (adding educational DVDs in the outpatient department and during admission) may also be helpful in other conditions, such as in cases of cardiac surgery or other treatment. In the future, we will conduct a study to evaluate the educational format on parental stress in parents of children undergoing surgical treatment for CHD. The COVID-19 pandemic will eventually be controlled and will disappear in the future. However, we may encounter another epidemic or pandemic. Our findings may inform future interventions for decreasing rejection rates for necessary healthcare as well as decreasing uncertainty and anxiety levels during admission in case of a similar global disaster. 

## Figures and Tables

**Figure 1 healthcare-11-01395-f001:**
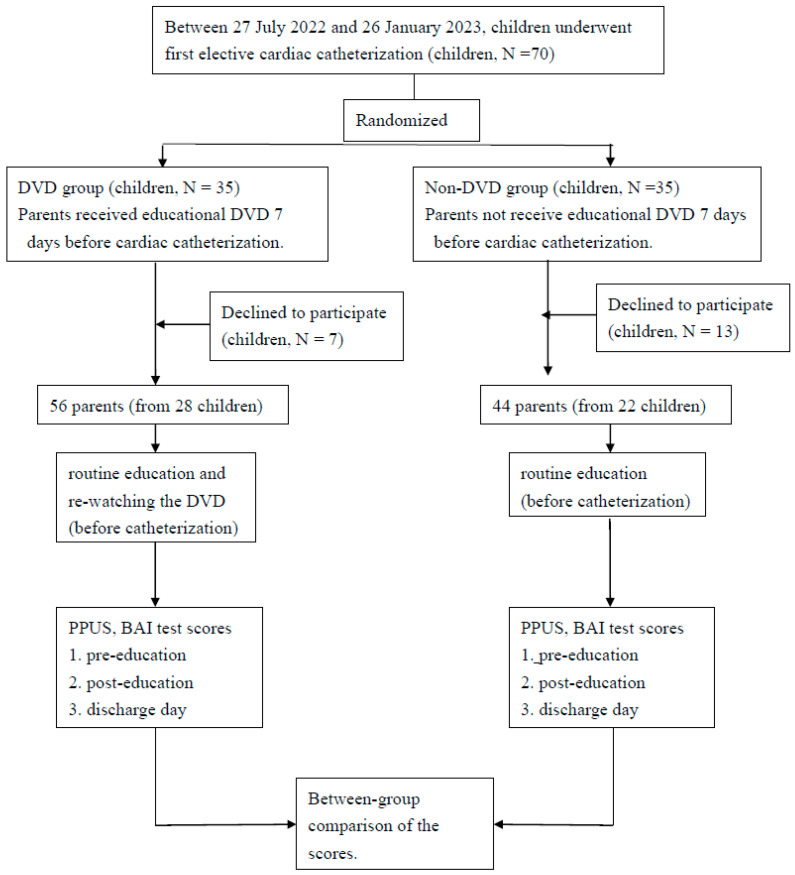
Flow diagram of the study.

**Table 1 healthcare-11-01395-t001:** Demographic data, between DVD group (n = 70) and non-DVD group (n = 70).

Group		DVD	Non-DVD	OR (95% CI)	*p*-Value
Parents’ Age *		31.5 ± 4.3	31.6 ± 4.9		0.962
Children’s age *		5.4 ± 2.1	5.6 ± 2.4		0.616
Parent’s education	low (%) high (%)	41 (58.6%) 29 (41.4%)	39 (55.7%) 31 (44.3%)	0.89 (0.46–1.87)	0.635
Family income	low (%) high (%)	32 (45.7%) 38 (54.3%)	30 (42.9%) 40 (57.1%)	0.89 (0.46–1.74)	0.762
Place of residence	urban (%) country (%)	43 (61.4%) 27 (38.6%)	46 (65.7%) 24 (34.3%)	1.20 (0.60–2.40)	0.514
Siblings	yes (%) no (%)	34 (48.6%) 36 (51.4%)	36 (51.4%) 34 (48.6%)	1.12 (0.58–2.18)	0.901
Children’s sex	male (%) female (%)	19 (54.3%) 16 (45.7%)	17 (48.6%) 18 (51.4%)	1.26 (0.65–2.44)	0.434
Children’s diagnosis	PDA or ASD (%) VSD (%)	16 (45.7%) 19 (54.3%)	15 (42.9%) 20 (57.1%)	1.12 (0.58–2.19)	0.650

DVD, digital versatile disk; PDA, patent ductus anteriosus; ASD, atrial septal defect; VSD, ventricular septal defect; OR, odds ratio; CI, confidence interval. * years; mean ± SD.

**Table 2 healthcare-11-01395-t002:** The rate of rejection for cardiac catheterization treatment between DVD group and non-DVD group in total and in different characteristics.

Group		DVD	Non-DVD	OR (95% CI)	*p*-Value
		Reject Number/All (%)	Reject Number/All (%)		
Total		14/70 (20.0%)	26/70 (37.1%)	0.42 (0.20–0.91)	0.025
Parental age	≤30 years ≥31 years	9/35 (25.7%) 5/35 (14.3%)	16/33 (48.5%) 10/37 (27.0%)	0.37 (0.13–1.02) 0.45 (0.14–1.48)	0.052 0.183
Parental education	low high	10/41 (24.4%) 4/29 (13.8%)	18/39 (46.2%) 8/31 (25.8%)	0.38 (0.15–0.97) 0.46 (0.12–1.73)	0.041 0.245
Family income	low high	6/32 (18.8%) 8/38 (21.1%)	12/30 (40.0%) 14/40 (35.0%)	0.35 (0.11–1.09) 0.50 (0.18–1.37)	0.065 0.171
Residence	urban country	8/43 (18.6%) 6/27 (22.2%)	13/46 (28.3%) 13/24 (54.2%)	0.59 (0.21–1.58) 0.24 (0.07–0.81)	0.284 0.019
Children’s siblings	no yes	8/36 (22.2%) 6/34 (17.6%)	16/34 (47.1%) 10/36 (27.8%)	0.32 (0.11–0.91) 0.56 (0.18–1.75)	0.029 0.313
Children’s sex	male female	10/38 (26.3%) 4/32 (12.5%)	14/34 (41.2%) 12/36 (33.3%)	0.51 (0.19–1.38) 0.29 (0.08–1.00)	0.182 0.043
Children’s diagnosis	PDA or ASD VSD	6/32 (18.8%) 8/38 (21.1%)	12/30 (40.0%) 14/40 (35.0%)	0.32 (0.11–1.09) 0.50 (0.18–1.37)	0.065 0.171
Children’s age	≤5 years ≥6 years	10/42 (23.8%) 4/28 (14.3%)	18/40 (45.0%) 8/30 (26.7%)	0.38 (0.15–0.98) 0.46 (0.12–1.74)	0.043 0.245

DVD, digital versatile disk; PDA, patent ductus anteriosus; ASD, atrial septal defect; VSD, ventricular septal defect; OR, odds ratio; CI, confidence interval.

**Table 3 healthcare-11-01395-t003:** Mishel’s Parent Perceptions of Uncertainty Scale (PPUS) between the DVD and non-DVD groups in different time points during admission.

	DVD (n = 56)	Non-DVD (n = 44)	Crude *p*-Value	Adjusted *p*-Value
Pre-education	128.3 ± 8.9	134.0 ± 7.3	0.001	<0.001
Post-education	99.0 ± 11.0	109.6 ± 12.2	<0.001	<0.001
Discharge day	64.2 ± 10.8	70.6 ± 11.0	0.004	0.008

Adjusted-*p* value: adjusted by parental age, sex, education; family income; place of residence; siblings or not; children’s age, sex, diagnosis of heart disease. DVD, digital versatile disk.

**Table 4 healthcare-11-01395-t004:** Beck Anxiety Inventory (BAI) scores between the DVD and non-DVD groups in different time points during admission.

	DVD (n = 56)	Non-DVD (n = 44)	Crude *p*-Value	Adjusted *p*-Value
Pre-education	20.6 ± 7.7	22.1 ± 8.6	0.361	0.464
Post-education	15.9 ± 7.3	19.3 ± 7.0	0.022	0.048
Discharge day	8.0 ± 3.8	9.7 ± 3.7	0.022	0.035

Adjusted-*p* value: adjusted by parental age, sex, education; family income; place of residence; siblings or not; children’s age, sex, diagnosis of heart disease. DVD, digital versatile disk.

## Data Availability

The data used to support the findings of this study are available from the corresponding author upon request.

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
