# Peer review of "Educational Videos to Reduce Parental Rejection of Pediatric Cardiac Catheterization during the COVID-19 Pandemic"

_healthcare, 2023, doi:10.3390/healthcare11101395_

Round 1
Reviewer 1 Report
The introduction should definitely be expanded and explained in more detail. The main idea of the study should be presented clearly and in detail in the text. The graphics and figures in this work should be more understandable. The conclusion part of the study should be explained in more detail. In addition, information about future studies in this field should be given with examples. The number of sources should be increased and new additions should be made to the number of current sources.
Author Response
Dear reviewer:
The revised manuscript entitled “Educational videos to reduce parental rejection of pediatric cardiac catheterization during the COVID-19 pandemic” has been revised in response to reviewer’s suggestions. The changes within the revised manuscript have been highlighted by underline.
Reviewer
Comment 1
The introduction should definitely be expanded and explained in more detail.
Response
Thank you for your comment and suggestion.
We have revised the third and fourth paragraphs of the Introduction to expand and explain the study. The revisions are indicated underlined in the revised manuscript.
【1.Introduction, paragraph 3】
【1.Introduction, paragraph 4】
Comment 2
The main idea of the study should be presented clearly and in detail in the text.
Response
Thank you for your comment and suggestion.
The main idea of the study is presented clearly and in detail in the paragraph 4. The revisions are indicated underlined in the revised manuscript.
【1.Introduction, paragraph 4】
Comment 3
The graphics and figures in this work should be more understandable.
Response
Thank you for your comment and suggestion.
We have revised the figures and tables.
【Figure, Table】
Comment 4
The conclusion part of the study should be explained in more detail.
Response
Thank you for your comment and suggestion.
We have re-written the “Implications for practice” subsection of the Discussion and the Conclusion section. The revisions are indicated underlined in the revised manuscript.
【4. Discussion: 4.2 Implications for practice, 5.Conclusions】
【5.Conclusions】
Comment 5
In addition, information about future studies in this field should be given with examples. The number of sources should be increased and new additions should be made to the number of current sources.
Response
Thank you for your comment and suggestion.
In conclusion, we have added the following part to the revised manuscript:
“Except for cardiac catheterization, the education format (adding educational DVD in the outpatient department and during admission) may also be helpful in other conditions, such as in cases of cardiac surgery or other treatment. In the future, we will conduct a study to evaluate the education format on parental stress in parents of children undergoing surgical treatment for CHD. The COVID-19 pandemic will be eventually controlled and will disappear in the future. However, we may encounter another epidemic or pandemic.”
【5.Conclusions: line 5-10】

Reviewer 2 Report
This manuscript addresses a very important topic for healthcare providers. I have the following suggestions/comments:
1. The introduction is relatively short. I believe that adding more information about the effectiveness of DVD's would be helpful to strengthen the authors' argument and further address the need for the current study.
2. How did the authors determine the required sample size?
3. Please add details about the setting(s) where this study was conducted?
4. Who delivered the routine education to the participants?
5. The authors used "Low" and "High" to categorize parents' income and education. However, they did not define what they mean by low and high.
6. Some p values are reported as 0.000 and they should be replaced with < 0.001.
7. It would be helpful to add more information in the implications section. Also, using single-sentence paragraph is not recommended in scholarly writing.
Good luck!
Author Response
Dear reviewer:
The revised manuscript entitled “Educational videos to reduce parental rejection of pediatric cardiac catheterization during the COVID-19 pandemic” has been revised in response to reviewer’s suggestions. The changes within the revised manuscript have been highlighted by underline.
Reviewer
This manuscript addresses a very important topic for healthcare providers. I have the following suggestions/comments:
Comment 1
The introduction is relatively short. I believe that adding more information about the effectiveness of DVD's would be helpful to strengthen the authors' argument and further address the need for the current study.
Response
Thank you for your comment and suggestion.
We have added the effectiveness of DVD's to the revised manuscript (Introduction, paragraph 3). The revisions are indicated underlined in the revised manuscript.
【Introduction, paragraph 3, line 2-12】
Comment 2
How did the authors determine the required sample size?
Response
Thank you for your comment and suggestion.
According to the study by Cohen**, G-power software was used to calculate the sample size. The parameter was: Effect size = 0.3; a = 0.05; Power = 0.8. The result was: 64 cases in each group. In our study, each group contains 70 cases (larger than 64 cases).
** Cohen, j. (1988). Statistical power analysis for the behavioral sciences (2nd ed.). Hillsdale, N. J. : L, Erlbaum Associates.
Comment 3
Please add details about the setting(s) where this study was conducted?
Response
Thank you for your comment and suggestion.
(1). In the outpatient department, we write
“Viewing of the DVD and the brief routine information were carried out in the outpatient department, with only the parents, child, physicians, and nurses who arranged the study.”
【2. Materials and Methods: 2.3 Group allocation and education in the outpatient department: line 7-9】
(2). After admission, we write
“After admission, viewing of the DVD, routine education, and BAI and PPUS tests were carried out in the education room of the pediatric ICU of the hospital. Only the parents, child, physicians, and nurses who arranged the study participated in the course.”
【2. Materials and Methods: 2.6 Instruments: line 6-9】
Comment 4
Who delivered the routine education to the participants?
Response
Thank you for your comment and suggestion.
We have added the following part to the revised manuscript:
“Experienced and trained physicians and nurses of the same team, who arranged the study, delivered the information verbally, obtained parents’ consent, and recorded the data. They also delivered the routine education and offered the DVD to the parents. The nurses and physicians who actually cared for the patients during admission were not involved in administering the educational protocol and were completely blinded to the study.”
【2. Materials and Methods: 2.5 Education after admission: line 5-9】
Comment 5
The authors used "Low" and "High" to categorize parents' income and education. However, they did not define what they mean by low and high.
Response
Thank you for your comment and suggestion.
We have added the following part to the revised manuscript:
“These confounding factors included parental age, sex, education (high: equal or higher than university level; low: equal or lower than junior college level), family income (high: family income more than USD 30,000 per year; low: family income equal or less than USD 30,000 per year), place of residence (urban or rural), presence of siblings, children’s age, sex, and heart disease type (PDA, ASD, or VSD).”
【2. Materials and Methods: 2.8 Confounding factors: line 2-5】
Comment 6
Some p values are reported as 0.000 and they should be replaced with < 0.001.
Response
Thank you for your comment and suggestion.
In the text and the tables, 0.000 was changed to <0.001.
Comment 7
It would be helpful to add more information in the implications section. Also, using single-sentence paragraph is not recommended in scholarly writing.
Response
Thank you for your comment and suggestion.
We have revised the implications section. The revisions are indicated underlined in the revised manuscript.
【4. Discussion: 4.2 Implications for practice】

Reviewer 3 Report
Thank you for the opportunity to review this interesting manuscript. The authors conducted a randomized-control study to compare the effectiveness of DVDs on parental rejection of pediatric cardiac catheterization during COVID-19 pandemic. The authors are to be congratulated on their achievements on this issue.
In my opinion, the authors provided an interesting and sound report. The objectives were clearly stated. The study method was adequately described. The results clearly presented. The discussion pointed out the important findings. The conclusions appropriately based on the results and discussions.
However, some concerns had raised from this work. Is the same team of doctors and nurses to deliver the information verbally and consent the parents? Who collected and recorded the data? Were they also blind to the study? Moreover, the result of multivariate regression analysis was not presented. I am wondering whether parents have adequate knowledge to decrease their anxiety. I am appreciated if the authors may discuss more about it. I also concern whether there are other potential biases from the study, and am appreciated if the authors may add it into the discussion section.
Line 53-54, the grammar might need to be verified.
Author Response
Dear reviewer:
The revised manuscript entitled “Educational videos to reduce parental rejection of pediatric cardiac catheterization during the COVID-19 pandemic” has been revised in response to reviewer’s suggestions. The changes within the revised manuscript have been highlighted by underline.
Reviewer
Thank you for the opportunity to review this interesting manuscript. The authors conducted a randomized-control study to compare the effectiveness of DVDs on parental rejection of pediatric cardiac catheterization during COVID-19 pandemic. The authors are to be congratulated on their achievements on this issue.
In my opinion, the authors provided an interesting and sound report. The objectives were clearly stated. The study method was adequately described. The results clearly presented. The discussion pointed out the important findings. The conclusions appropriately based on the results and discussions.
However, some concerns had raised from this work.
Comment 1
Is the same team of doctors and nurses to deliver the information verbally and consent the parents?
Who collected and recorded the data? Were they also blind to the study?
Response
Thank you for your comment and suggestion.
We have added the following part to the revised manuscript:
“Experienced and trained physicians and nurses of the same team, who arranged the study, delivered the information verbally, obtained parents’ consent, and recorded the data. They also delivered the routine education and offered the DVD to the parents. The nurses and physicians who actually cared for the patients during admission were not involved in administering the educational protocol and were completely blinded to the study.”
【2. Materials and Methods: 2.5 Education after admission: line 5-9】
Comment 2
Moreover, the result of multivariate regression analysis was not presented.
Response
Thank you for your comment and suggestion.
The confounding factors were adjusted using multivariate regression analysis. Moreover, the p-values adjusted by the confounding factors (adjusted p-value) are presented in Tables 3 and 4.
Comment 3
I am wondering whether parents have adequate knowledge to decrease their anxiety. I am appreciated if the authors may discuss more about it.
Response
Thank you for your comment and suggestion.
We have added a new paragraph in the revised manuscript to describe this question.
“During the outbreak of the COVID-19 pandemic, the tsunami of the ………….
【4. Discussion: Sixth paragraph】
Comment 4
I also concern whether there are other potential biases from the study, and am appreciated if the authors may add it into the discussion section.
Response
Thank you for your comment and suggestion.
We re-write the Discussion, Limitations part, and add some bias.
【4. Discussion: 4.1 Limitations】
Comment 5
Line 53-54, the grammar might need to be verified.
Response
Thank you for your comment and suggestion.
have revised this sentence as follows:
“A single blind, randomized case-control study was conducted.”
【2. Materials and Methods: 2.1 Design, line 1-2】

Reviewer 4 Report
The authors investigated educational DVDs' efficacy in reducing parental rejection of pediatric cardiac catheterization during the COVID-19 pandemic. The authors concluded that pre-admission DVDs decrease uncertainty and consequently rejection of catheterization. The effect was enforced in the parents with lower education, rural residence, with only one child, female child, or younger child. Moreover, they found that re-watching DVDs during admission also reduces parental uncertainty and anxiety.
The reviewer almost agrees with their research and conclusion, with only a few comments.
Comment 1: Did the catheter procedures in this study include an only case with catheter treatment for the patient's cardiac defect, or also include a case with only a catheter study before surgical correction? If the latter, I think that whether the catheter procedure includes interventional treatment makes the difference in the parent's uncertainty or anxiety. The author should clearly state the content of the catheter procedure and also should mention the effect of whether or not catheterization includes treatment in the manuscript, if necessary.
Author Response
Dear reviewer:
The revised manuscript entitled “Educational videos to reduce parental rejection of pediatric cardiac catheterization during the COVID-19 pandemic” has been revised in response to reviewer’s suggestions. The changes within the revised manuscript have been highlighted by underline.
Reviewer
The authors investigated educational DVDs' efficacy in reducing parental rejection of pediatric cardiac catheterization during the COVID-19 pandemic. The authors concluded that pre-admission DVDs decrease uncertainty and consequently rejection of catheterization. The effect was enforced in the parents with lower education, rural residence, with only one child, female child, or younger child. Moreover, they found that re-watching DVDs during admission also reduces parental uncertainty and anxiety.
The reviewer almost agrees with their research and conclusion, with only a few comments.
Comment 1:
Did the catheter procedures in this study include an only case with catheter treatment for the patient's cardiac defect, or also include a case with only a catheter study before surgical correction? If the latter, I think that whether the catheter procedure includes interventional treatment makes the difference in the parent's uncertainty or anxiety. The author should clearly state the content of the catheter procedure and also should mention the effect of whether or not catheterization includes treatment in the manuscript, if necessary.
Response
Thank you for your comment and suggestion.
In this study, only case with catheter treatment for the patient's cardiac defect were included in the study. We have added the following part to the revised manuscript:
“Between July 27, 2022, and January 26, 2023, 70 children aged 3–12 years underwent elective and necessary cardiac catheterization treatment for patent ductus arteriosus (PDA), atrial septal defect (ASD), or ventricular septal defect (VSD). The treatment included trans-catheter closure of PDA, trans-catheter closure of ASD, and trans-catheter closure of VSD.”
【2. Materials and Methods: 2.2 Participants: line 1-4】

Round 2
Reviewer 2 Report
Thanks for addressing my comments and suggestions.